# The Hospitality Industry in the Face of the COVID-19 Pandemic: Current Topics and Research Methods

**DOI:** 10.3390/ijerph17207366

**Published:** 2020-10-09

**Authors:** Mohammad Reza Davahli, Waldemar Karwowski, Sevil Sonmez, Yorghos Apostolopoulos

**Affiliations:** 1Department of Industrial Engineering and Management Systems, University of Central Florida, 12800 Pegasus Dr., Orlando, FL 32816, USA; wkar@ucf.edu; 2College of Business Administration, University of Central Florida, 12744 Pegasus Dr., Orlando, FL 32816, USA; sevil@ucf.edu; 3Complexity & Computational Population Health Group, Texas A&M University, 2929 Research Pkwy, College Station, TX 77845, USA; yaposto@tamu.edu

**Keywords:** COVID-19 pandemic, hospitality industry, impact, tourism, systematic review

## Abstract

This study reports on a systematic review of the published literature used to reveal the current research investigating the hospitality industry in the face of the COVID-19 pandemic. The presented review identified relevant papers using Google Scholar, Web of Science, and Science Direct databases. Of the 175 articles found, 50 papers met the predefined inclusion criteria. The included papers were classified concerning the following dimensions: the source of publication, hospitality industry domain, and methodology. The reviewed articles focused on different aspects of the hospitality industry, including hospitality workers’ issues, loss of jobs, revenue impact, the COVID-19 spreading patterns in the industry, market demand, prospects for recovery of the hospitality industry, safety and health, travel behavior, and preference of customers. The results revealed a variety of research approaches that have been used to investigate the hospitality industry at the time of the pandemic. The reported approaches include simulation and scenario modeling for discovering the COVID-19 spreading patterns, field surveys, secondary data analysis, discussing the resumption of activities during and after the pandemic, comparing the COVID-19 pandemic with previous public health crises, and measuring the impact of the pandemic in terms of economics.

## 1. Introduction

On December 8, 2019, the government of Wuhan, China, announced that health authorities were treating dozens of new virus cases, identified as coronavirus disease 2019 (COVID-19) [1]. Since then, COVID-19, a new strain of SARS (SARS-CoV-2), has grown into a global pandemic and spreading across many countries. A highly transmissible respiratory disease, COVID-19 spreads through contact with other infected individuals, with symptoms such as fever, cough, and breathing problems [2]. Transmission can also occur from asymptomatic individuals, with up to 40% of infected persons remaining asymptomatic [3]. Other factors that facilitate infection include (1) speed and efficiency of COVID-19 transmission; (2) airborne transmission [4]; (3) close contact between infected and non-infected individuals; (4) vulnerability of immunocompromised individuals with specific underlying health conditions (e.g., hypertension, diabetes, cardiovascular disease, respiratory problems); (5) susceptibility of persons over 65; and (7) contact with persons who have traveled to locations with a high number of cases [5].

Critical global responses to control the spreading of the COVID-19 pandemic have included travel restrictions, shelter-in-place and social distancing orders. Most countries around the world have imposed partial or complete border closures, with travel bans affecting the majority of the world’s population [6]. With millions suddenly unemployed, uncertainty over economic recovery, and global fears of continuing COVID-19 spread and its future waves, the hospitality industry was among the first industries affected, and it will be among the last industries to recover [7].

On 20 January 2020, the United States reported its first COVID-19 confirmed case [8]. In February and through March 2020, the pandemic began to exact unprecedented economic and social consequences. Since public health concerns started to escalate in mid-February 2020, U.S. hotels have lost room revenues [9,10]. As of 3 June 2020, six out of ten hotel rooms remain empty across the country [11,12]. Since August 2020, almost half of the hotel industry employees are still not working, and five out of ten rooms are empty [10].

The present study focuses on understanding the current research on the hospitality industry’s topic in the face of the COVID-19 pandemic. A systematic review of the contemporary literature is considered to identify and classify research that focuses on the hospitality industry in the time of COVID-19. The systematic review’s primary purpose is to identify, summarize, and analyze the findings of all relevant individual studies that are addressing predefined research questions [13].

Although no study used a systematic literature review to investigate the hospitality industry in the face of the COVID-19 pandemic, conducting the systematic review is common in the context of hospitality. For example, Yu et al. [14] conducted a comprehensive review of abusive supervision in hospitality and tourism. Gorska-Warsewicz and Kulykovets [15] conducted a systematic literature review and selected 26 studies to analyze hotel brand loyalty. The study used the Joanna Briggs Institute’s critical appraisal checklist to address the risk of bias among the included records. Sharma et al. [16] used a systematic review and analyzed 403 published papers in 13 established hospitality journals to address green hospitality practices. The study proposed a unified conceptual framework based on discovering seven research areas under eco-innovative procedures. Chi et al. [17] discussed applying artificial intelligence in the hospitality industry, specifically service delivery. The study reviewed 63 publications and identified seven major themes.

Regarding the COVID-19 pandemic, many researchers used a systematic literature review to summarize and evaluate the results of all relevant studies. For example, de Pablo et al. [18] presented a systematic review of physical and mental health outcomes in health care workers exposed to COVID-19. The study reviewed 115 grey literature publications and published papers in Web of Science until 15th April 2020. Luo et al. [19] conducted a systematic review using Google Scholar, PubMed, Embase, and WHO COVID-19 databases on the psychological and mental impact of COVID-19 among the general population, healthcare workers, and patients with higher COVID-19 risk. The study selected sixty-two studies with 162,639 participants from seventeen countries.

For the present study, the preferred reporting items for systematic reviews and meta-analyses (PRISMA) are considered for ensuring reliable and meaningful results of the systematic literature review studies. The PRISMA protocol consists of 27 items that help researchers prepare and report scientific evidence accurately and reliably, which improves the quality of research [13]. This review is structured as follows: the methodology section discusses inclusion and exclusion criteria and the risk of bias; the results, research approaches used, and discussion sections provide outputs of the literature search and describes the status of the hospitality industry at the time of COVID-19.

## 2. Methodology

The literature review follows Preferred Reporting Items for Systematic Reviews and Meta-Analyses (PRISMA) guidelines [13,20] and contains two main features: developing research questions and determining search strategy. The following research questions have guided this review:

RQ1. What aspects of the hospitality industry at the time of the COVID-19 pandemic have been studied?

RQ2. What research methodologies have been used to investigate the impact of COVID-19 on the hospitality industry?

In order to address the above questions, a search strategy was developed to list and review all relevant scientific papers by (a) defining keywords and identifying all relevant materials, (b) filtering the identified records, and (c) addressing the risk of any bias [13]. One of the main steps in a systematic review is developing specific keywords. Herein, our objective was to target all critical segments of the hospitality industry (e.g., hotels, restaurants) and the broadly defined tourism industry. The defined keywords are shown in Table 1.

Web of Science, Science Direct, and Google Scholar were used as database search tools. Keywords were used to discover relevant articles and identify 175 articles with relevant content. Because this topic is rapidly evolving, it is important to mention that article discovery was finished at the end of August 2020. After developing the main database and identifying all relevant papers, a formal screening process based on specific exclusion and inclusion criteria was followed. Because of the very timely issue of the COVID-19 pandemic, we included documents in the forms of peer-reviewed academic publications, grey literature, and pre-print articles. However, we excluded secondary sources that were not free or open access, letters, newspaper articles, viewpoints, presentations, anecdotes, and posters.

The screening of the titles, abstracts, conclusions, and keywords in the identified records after removing duplication (*n* = 168) resulted in excluding articles (*n* = 115) because of not enough relating to the topic. The remaining articles (*n* = 53) were read in full against the eligibility principle, and three articles were excluded for not addressing the research questions.

Selection bias in a systematic review can occur by the erroneous application of inclusion/exclusion criteria and/or the specification of included papers’ dimensions. To address the first type of bias, two researchers (MD and WK) independently reviewed the title, abstract, and conclusions of the identified records to select articles for the full-text review. Subsequently, the two researchers compared their selected articles to reach a consensus. After reading the full text of the selected papers, the authors decided whether to include the article—which was considered and included upon reaching an agreement. Disagreements were resolved by the input of the other two authors (S.S. and Y.A.). To address the second type of bias, two researchers (MD and WK) independently specified the included papers’ classifications and subsequently compared the results, resolving disagreements by consultation with the other authors (S.S. and Y.A.). The selection strategy, as per PRISMA guidelines, is illustrated in Figure 1.

## 3. Results

All included articles were categorized and stored in the main database according to year, source of publication, the industry segment, geographic location, research approach, aspect of the hospitality industry, and methodology. The characteristics of the included papers are shown in Table 2.

The publication sources of the included papers are illustrated in Figure 2. The most popular publication sources include *Tourism Geographies*, *International Journal of Infection Diseases*, and *Journal of Tourism and Hospitality Education*.

To generate a better picture of the included papers, the map of the co-occurrence of terms in the title and abstract is shown in Figure 3. The colorful nodes are associated with specific terms, and their sizes represent the frequency of term occurrence, and links between two nodes indicate the co-occurrence of the terms. In this Figure, frequently co-occurring terms create clusters and appear closer with the same color. A first glance at Figure 3 reveals the central cluster (blue color) with terms including COVID-19, health, travel, effect, and global tourism.

Fifteen papers investigated the hospitality industry in the face of COVID-19 on the global scale, as shown in Figure 4. Other articles focused on a specific country or location such as China (nine papers), India (six papers), or the United States (four papers).

The included papers used different research approaches to investigate the impact of COVID-19 on the hospitality industry (see Figure 5). Each approach is explained in the following section.

## 4. Research Approaches Used

The reviewed papers used different research approaches and focused on various subjects related to the hospitality industry during the COVID-19 pandemic. However, all papers have been classified into six groups as follows: (1) developing simulation and scenario modeling, (2) reporting impacts of the COVID-19 pandemic, (3) comparing the COVID-19 pandemic with previous public health crises, (4) measuring impacts of the COVID-19 pandemic in terms of economics, (5) discussing the resumption of activities during and after the pandemic, and (6) conducting surveys. Since some of the reviewed papers belong to more than one group, these have been assigned to the dominant group.

### 4.1. Developing Simulation & Scenario Modeling

Eight included papers in this review applied simulation & scenario modeling to estimate aspects of tourism demand and the COVID-19 spreading pattern. The studies used different models and analyses, including a dynamic stochastic general equilibrium (DSGE) model, supply and demand curve, agent-based model, epidemiological model, and susceptible exposed infected recovered (SEIR) model.

Yang et al. [2] applied DSGE, a macroeconomics technique that depicts economic phenomena based on the general equilibrium framework, to investigate the impacts of increasing health disaster risk (the pandemic) and its persistence on the model parameters such as tourism demand. He incorporated two indicators (health status, and health disaster) and three categories of decision-makers (the government, households, and producers) into the DSGE model concerning the tourism sector. The findings are not surprising and point out that the longer pandemic will have a more devastating effect on the hospitality industry.

Bakar and Rosbi [1] utilized a supply and demand curve to analyze the economic impact of COVID-19 on the hospitality industry. In order to develop the supply and demand curve, the demand function was created by using factors of *price setting of selected goods, tastes and preferences of customers, customers’ expectations, the average income of certain countries, and the number of buyers*. Meantime, the supply function is developed by using elements of *production techniques, resource price, price expectations, price of related goods, supply stocks, and numbers of sellers*. The supply and demand curve was then developed in the market equilibrium condition where the demand in the market is equal to the supply in the market. Finally, changes in market equilibrium as the result of the COVID-19 outbreak were investigated. The results indicate that the pandemic created some "panic" level among people and consequently decreased overall demand in the tourism and hospitality industry [1]. The study urged governments to discover a vaccine as quickly as possible and identify policies to prevent the further decrease in demand for tourism and hospitality services during the post-pandemic period [1].

D’Orazio et al. [38] used an agent-based model to determine the virus spreading in tourist-oriented cities and, consequently, discover sustainable and resilient strategies [38]. The model represented simulated individuals’ movement and the contagion virus spreading approach (the epidemic rules based on previous studies) in a touristic urban area. The model calculated the probability that an infector: (i) could infect a susceptible individual j based on a linear combination of the current incubation time of (i), the exposure time, and the mask filter adopted by both i and j. The model evaluated the number of infectors within the touristic urban area over time and the number of visitors who return home being infected over time. After analyzing different scenarios, such as “social distancing-based measures” and “facial mask implementation”, the results reveal that “social distancing-based measures” were related to significant economic losses [38]. This phenomenon appears to be an effective policy in locations with the highest infection rates [38]. However, “social distancing-based measures” lose their advantage in areas of low infection rates and a high degree of "facial mask implementation" [38].

Five studies investigated COVID-19 cases and spreading patterns on the Diamond Princess cruise ship. On February 1, 2020, a disembarked passenger from the ship tested positive for COVID-19 [69], after which the 3711 passengers were quarantined [69]. By the end of the quarantine, more than 700 passengers were infected with COVID-19 [69]. Fang et al. [33] developed the flow of passengers (crowd flow simulation model) on the Diamond Princess cruise ship, and then created the virus transmission rule between individuals to simulate the spread of the COVID-19 caused by the close contact during passengers’ activities. Mizumoto and Chowell [34] and Mizumoto et al. [35] developed an epidemiological model based on discrete-time integral equations and daily incidence series. Rocklöv et al. [36] collected data on confirmed cases on the Diamond Princess cruise ship. They used the SEIR model (compartmental technique estimating the number of susceptible (S), exposed (E), infected (I), and recovered (R) individuals) to calculate the primary reproduction number. The basic reproduction number is *the expected number of cases directly generated by one case in a population where all individuals are susceptible to infection* [70]. Zhang et al. [37] collected data of daily incidence for COVID-19 on the Diamond Princess cruise ship, data of a serial interval distribution (*the time between successive cases in a chain of transmission* [71]), and applied "projections" package in R to calculate the basic reproduction number. The studies concluded that the cruise company’s immediate response in following recommended safety guidelines and early evacuation of all passengers could prevent mass transmission of COVID-19 [33,34,35,36,37].

### 4.2. Reporting the Impacts of the COVID-19 Pandemic

Seventeen papers applied secondary data analysis to report COVID-19 pandemic’s impacts on the hospitality industry. Because of the ongoing pandemic and publication time of included papers, secondary data sources have been invaluable for most studies in this review. The studies reported impacts of the pandemic on different aspects of the hospitality industry, including job loss, revenue losses, access to loans, market demand, emerging new markets, hostile behaviors towards foreigners, and issues of hospitality workers and hotel cleaners.

Nicola et al. [49] summarized the pandemic’s impact on the global economy by reviewing news distributed by mass-media, government reports, and published papers. To better understand the impacts of the pandemic, the study divided the world economy into three sectors of primary (including agriculture, and petroleum & oil), secondary (including manufacturing industry), and tertiary (including education, finance industry, healthcare, hospitality tourism and aviation, real estate, sports industry, information technology, and food sector). They reported job loss, revenue losses, and decreasing market demand in the hospitality, tourism, and aviation sectors [49].

Ozili and Arun [51] provided a list of COVID-19 statistics, including confirmed cases, confirmed deaths, recovered cases in several countries and continents, and discussed the global impact of COVID-19 on the travel and restaurant industries. The study reviewed different policy measures implemented by different countries around the world to deal with COVID-19. Ozili and Arun [51] categorized these into four groups of (1) human control measures; (2) public health measures; (3) fiscal measures; and (4) monetary measures. In the human control policies measures, different actions including foreign travel restrictions, internal travel restrictions, state of emergency declarations, limiting mass gathering, closing down of schools, and restricting shops and restaurants, have also been identified [51].

Several studies reported the effect of COVID-19 on specific critical domains of the hospitality industry, such as undocumented workers and hotel cleaners. Williams and Kayaoglu argued that the most vulnerable workers in the industry need governmental financial support but cannot receive assistance, most likely because they are undocumented immigrants [55]. Furthermore, Rosemberg [50] highlighted the issues of job insecurity, risk of exposure to COVID-19, lack of health insurance, added pressure due to increased workload, and extra time required for ensuring complete disinfection during the pandemic [50].

Other studies focused on the pandemic’s impacts on specific countries, including China, Malaysia, Nepal, and India. Wen et al. [54] reviewed literature and news on Chinese tourist behavior, tourism marketing, and tourism management; they concluded the growing popularity of luxury trips, free and independent travel, and medical and wellness tourism post-COVID-19 period [54]. They indicated that new forms of tourism would be more prevalent in post-COVID-19, including (1) slow tourism, which emphasizes local destinations and longer lengths of stay, and (2) SMART tourism, which uses data analytics to improve tourists’ experiences [54]. Another study used automated content analysis to investigate newspaper articles and identified nine key themes among 499 newspaper articles, including, “*COVID-19’s impact on tourism, public sentiment, the role of the hospitality industry, control of tourism activities and cultural venues, tourism disputes and solutions, national command and local response, government assistance, corporate self-improvement strategies, and post-crisis tourism product*” [42].

### 4.3. Comparing COVID-19 with Previous Public Health Crises

Two papers compared the COVID-19 pandemic with previous public health crises. In the first study, lessons learned from previous crises and pandemics are discussed, including malaria, yellow fever, Ebola, Zika virus, Middle East respiratory syndrome (MERS-CoV), avian influenza (H5N1), Creutzfeldt-Jakob disease (Mad Cow disease), swine flu (H1N1), and severe acute respiratory syndrome (SARS) [22]. This paper concluded that the impacts of COVID-19 on the global economy and China’s tourism and hospitality industry, in particular, are likely to differ from previous pandemics, from which the tourism and hospitality industry recovered relatively quickly [22].

Gössling et al. [21] reviewed the impact of previous crises on global tourism, including the Middle East Respiratory Syndrome (MERS) outbreak (2015), the global economic crisis (2008–2009), the SARS outbreak (2003), and the September 11 terrorist attacks (2001) [21]. The authors indicated that previous crises did not have long-term impacts on global tourism. The authors also warned about increasing pandemic threats for several reasons, including the fast-growing world population, rapidly developing global public transportation systems, and increasing consumption of processed/low-nutrition foods [21]. Gössling et al. [21] also discussed the impact of COVID-19 on different hospitality industry sectors. The authors distinguished the impact of COVID-19 in view of two different aspects of (1) observed impacts (e.g., declines in hotel occupancy rates, liquidity problems in the restaurant industry); and (2) projected impacts (e.g., revenue forecasts in the accommodations sector, estimation of revenues) [21].

The still-evolving understanding of the coronavirus’s behavior makes it difficult to predict the industry’s recovery in the near future. However, suggestions have already been made for post-COVID-19 management of the tourism and hospitality industry. These include: (1) focusing primarily on domestic tourism; (2) ending mass tourism and pilgrimage tourism; (3) focusing more on conference tourism, virtual reality tourism, and medical tourism; and (4) building a more sustainable tourism and hospitality industry rather than a return to "business as usual" [21,22].

### 4.4. Measuring the Impacts of COVID-19 in Terms of Economics

Five papers measured the impacts of the pandemic on the hospitality industry in terms of economics. These studies used different models and analyses, including seasonal autoregressive integrated moving average model, scenario analysis, and trend analysis. The economic impact of COVID-19 on the tourism and hospitality industry has been examined in terms of lost earnings or jobs. Centeno and Marquez [29] developed seasonal autoregressive integrated moving average models for the Philippines’ tourism and hospitality industry, forecasting the total earnings loss of around 170.5 billion PHP (Philippine Peso)—equivalent to $3.37 billion—from COVID-19 just until the end of July 2020. To ease the pandemic’s effects on the hospitality industry, the authors propose dividing the country into two regions according to the level of infection risk (high-risk and low-risk of COVID-19) to allow domestic travel into low-risk regions or areas [29].

Günay et al. [30] applied a scenario analysis technique to calculate the impact of COVID-19 on Turkey’s tourism and hospitality industry. Their model predicts the total loss of revenues in the best and the worst scenarios as $1.5 billion and $15.2 billion, respectively, for 2020 [30]. The worst-case scenario involves the closing of borders for four months without any economic recovery [30]. The authors indicated that this would be one of Turkey’s worst tourism crises under the worst-case scenario, exceeding the losses from public health crises due to Swine flu, Avian Flu, and SARS [30].

Mehta [31] estimated the effect of COVID-19 on India’s economy at an earnings loss of about $28 billion in 2020, along with 70% job losses for tourism and hospitality workers, and mass bankruptcies [31]. Trend analysis was also used to examine the impact of COVID-19 on the global tourism and hospitality industry and global GDP [32]. According to Priyadarshini [32], the real global GDP growth will drop from 2.9% in 2019 to 2.4% by the end of 2020, while global revenues for the tourism and hospitality industry will drop by 17% compared to 2019. The study also predicts that North America, Europe, and Asia will experience the most massive losses in global revenues. The tourism and hospitality revenues will fall in the U.S., Germany, Italy, and China by 10%, 10%, 24%, and 40%, respectively [32].

Cajner et al. [28] analyzed the COVID-19 pandemic impact on the U.S. labor market. The study calculated that about 13 million paid jobs were lost between March 14 to 28, 2020. To better understand this number’s significance, the authors pointed out that only nine million private payroll employment jobs were lost during the Great Recession of the 1930s (less than 70% of the pandemic job loss) [28]. The study also highlighted that the leisure and hospitality industry was the hardest hit and most affected industrial sector [28].

### 4.5. Resumption of Activities during and after the Pandemic

Thirteen papers recommended various remedial and management actions for the resumption of activities during and after the pandemic. The consequences of COVID-19 on the hospitality industry, such as empty hotels and loss of jobs, are discussed in one paper that offers a positive outlook that the industry will receive a significant flow of guests upon the easing of travel bans and restrictions [63]. The author stressed the importance of support for the hospitality industry during the pandemic and the need for proper guidance to ensure successful reopening during the post-pandemic period. Taking a different perspective, another study suggests that the hospitality industry may not do well after the lifting of travel bans and mobility restrictions [58]. The study refers to a survey that found more than half of the participants would not order food even after the pandemic ends. The author also recommends a series of actions for restaurants to attract customers in the post-COVID-19 period, such as including island-sitting arrangements to assure maximum physical distances between people, live cooking counters to allow customers to watch their food being prepared to instill confidence in its safety, and having appropriate hygiene and cleaning procedures throughout [58].

Bagnera et al [68] investigated the impact of COVID-19 on hotel operations and recommended a series of actions for hotel owners and managers, including using fewer rooms (reducing hotel capacity); emphasizing take-out or delivery options to reduce public dining, implementing intensified cleaning/sanitizing protocols; committing to the use of personal protective equipment (PPE) for workers and increasing attention to personal hygiene; communicating new COVID-19 policies to guests and employees; implementing physical distancing practices in public areas, and implementing protocols for guests exposed to or infected by COVID-19 [68]. It should be noted that the World Health Organization (WHO) produced a guide titled "Operational Considerations for COVID-19 Management in the Accommodations Sector" to provide practical assistance to the hospitality sector in particular [64]. The report is divided into sections for the management team, reception and concierge, technical and maintenance services, restaurants and dining rooms and bars, recreational areas for children, and cleaning and housekeeping with a list of responsibilities to help manage the threat of COVID-19 [64]. Furthermore, Jain discussed different hotel industry strategies to bring back customers, including disposable utensils in rooms, emphasizing staff health and hygiene, and using UV light to disinfect [59].

Specific steps for an exit strategy and the reopening of activities in different business sectors are presented by Peterson et al. [62]. Primary steps include implementing widespread COVID-19 testing, having enough PPE supply, lifting social distancing and mobility restrictions, using electronic surveillance, and implementing strategies to decrease workplace transmission [62]. Emphasis was placed on the daily screening of hospitality sector staff for COVID-19 by using real-time reverse transcription-polymerase chain reaction or serology tests [62]. In this aspect, another study used primary and secondary data and applied the descriptive analysis method to explore revitalization strategies for small and medium-sized businesses, especially in the tourism industry, after COVID-19 in Yogyakarta [56]. The study recommended several policies, such as implementing banks’ credit policies with simpler processes and lower interest [56].

Several papers discussed redesigning and transforming the tourism and hospitality industry after COVID_19 pandemic. The proposed ideas include increasing resilience and security of the tourism and hospitality workforce in post-COVID-19 by cross-training and teaching different skills to workers [61]; exploiting the unique opportunity presented by COVID-19 to transform and refocus the tourism and hospitality industry towards local attractions rather than global destinations, and redesigning spaces to assure a 6-foot distance between tourists [57,60,67]. Hao et al. [65] developed a COVID-19 management framework as a result of reviewing the overall impacts of the COVID-19 pandemic on China’s hotel industry. The framework contains three main elements of an anti-pandemic process, principles, and anti-pandemic strategies. The anti-pandemic process adopted the six phases of disaster management, including the pre-event phase (taking prerequisite actions), the prodromal phase (observing the warning signs), the emergency phase (taking urgent actions), the intermediate phase (bringing back essential community services), the recovery phase (taking self-healing measures), the resolution phase (restoring the routine).

Hao et al. [65] recommended four principles for the different phases of disaster management, including disaster assessment, ensuring employees’ safety, customer & property, self-saving, and activating & revitalizing business. Finally, the study discussed the main anti-pandemic strategies in the categories of leadership & communication, human resource, service provision, corporate social responsibility, finance, and standard operating procedure. Recently, Sönmez et al [66] reviewed the impacts of the COVID-19 pandemic on immigrant hospitality workers’ health and safety. The study indicated that while a significant rise in occupational stress has been observed in immigrant hospitality workers over the past 15–20 years, the COVID-19 pandemic can add more pressure on workers and potentially deteriorate their mental and physical health condition. The authors recommended different actions in aspects of public and corporate policy, workplace policy, and future research areas.

### 4.6. Conducting Surveys 

Five papers conducted survey studies to investigate different hospitality industry aspects, including social costs, customer preference, expected chance of survival, and travel behavior. Qiu et al. [25] developed the contingent valuation method to estimate costs borne by residents of tourist destinations (social costs) due to the COVID-19 pandemic. Contingent valuation is *a survey-based economic technique for the valuation of non-market resources* [72]. The survey asks questions about how much money residents would be willing to pay to keep a specific resource. The study attempted to investigate how residents perceive the risk of tourism during the COVID-19 pandemic. By considering three Chinese urban destinations, Qiu et al. [25] quantified tourism’s social costs during the pandemic. The results indicate that most residents were willing to pay for risk reduction, but this payment differs based on respondents’ age and income.

Alonso et al. [27] focused on the theory of resilience and conducted a survey from a sample of 45 small hospitality businesses to answer questions about participants’ main concerns regarding the COVID-19 pandemic. How small hospitality businesses are handling this disruption. Furthermore, what are the impacts of the pandemic on day-to-day activities. Alonso et al. [27] analyzed the qualitative responses through content analysis. The study highlighted nine theoretical dimensions about owners-managers’ actions and alternatives when confronted with the COVID-19 pandemic.

Kim and Lee [26] studied the impacts of the perceived threat of the COVID-19 pandemic on customers’ preference for private dining facilities. The study conducted a survey and concluded that the salience of the COVID-19 increases customers’ preference for private dining facilities.

Bartik et al. [23] discussed the impact of COVID-19 on the U.S. small businesses, especially restaurants and tourism attractions, and highlighted their fragile nature in the face of a prolonged crisis. Such companies typically have low cash flow, and in the face of this pandemic, they will either have to declare bankruptcy, take out loans, or significantly cut expenses [23]. Their restaurant owners’ survey found that the expected chance of survival during a crisis lasting one month is 72%, for a crisis that lasts four months is 30%, and for a crisis that lasts six months is 15%. The result also indicated that more than 70% of U.S. small businesses want to take up the CARES Act Paycheck Protection Program (PPP) loans, even though most of them believe it would be challenging to establish eligibility for receiving such loans [23].

Finally, a survey study by Nazneen et al. [24] investigated the pandemic’s impact on travel behavior and reported that it had significant impacts on tourists’ decisions to travel for the next 12 months. The authors also concluded that respondents are concerned about hotels’ safety and hygiene, recreational sites, and public transports [24]. It has also been postulated that hygiene and safety perception will play a significant role in travel decisions in post-COVID-19 times [24].

## 5. Discussion

Even though included papers studied different aspects of the hospitality industry during the COVID-19 pandemic (see Figure 6), the main topics relate to recovery of the industry (19% of papers), market demand (18% of papers), revenue losses (16% of papers), the COVID-19 spreading patterns in the industry (14% of papers), job losses (10% of papers), safety and health aspects (8% of papers), issues related to the employment of hospitality workforce (7% of papers), travel behaviors (4% of papers), preferences of customers (2% of papers), and social costs of pandemic (2% of papers).

The employment issues of hospitality workers have been mentioned by 7% of papers in the categories of “reporting the impacts of the COVID-19 pandemic” and “discussing the resumption of activities”. These papers discussed job insecurity, financial, and health issues among documented and undocumented workers. Ten percent of included papers reported or measured job losses in the hospitality industry as the result of the COVID-19 pandemic. Revenue losses, market demand, and recovery of the industry were the most popular aspects of the hospitality industry, and 16%, 18%, and 19% of the included papers, respectively, discussed these topics. It should be noted that these aspects were mainly discussed in the framework of “reporting the impacts of the COVID-19”. The aspect of COVID-19 spreading patterns was the most popular topic in the approach of “developing simulation & scenario modeling”. Eight percent of included papers recommended different safety actions for the resumption of activities during and after the pandemic. Travel behaviors, preferences of customers, and social costs were mainly analyzed in the “conducting surveys” approach.

The reviewed papers used a variety of research models and analyses to study the hospitality industry in the face of COVID-19 (see Figure 7). Secondary data analysis was utilized to study almost all aspects of the hospitality industry. COVID-19 spreading patterns were investigated by using several quantitative models, including the SEIR models, epidemiological models, agent-based models, and crowd flow simulation models. The seasonal autoregressive integrated moving average model was used to calculate job loss and revenue losses. The contingent valuation method, content analysis, and analyzing questionnaire data were parts of the “conducting surveys” approach and were used to analyze social and behavioral aspects of the hospitality industry response to the COVID-19 epidemic.

## 6. Conclusion

This paper provides a systematic review of the published research topics relevant to the understanding of the hospitality industry in the time of COVID-19 pandemic. By selecting keywords and following PRISMA guidelines, we explored two main research questions related to the objective. A total of 50 papers that met the predefined inclusion criteria were included in the review. The following two research questions have been explored:

RQ1. What aspects of the hospitality industry at the time of the COVID-19 pandemic have been studied?

RQ2. What research methodologies have been used to investigate the impact of COVID-19 on the hospitality industry?

The included papers were classified into six thematic groups, including: (1) developing simulation and scenario modeling, (2) conducting surveys, (3) reporting impacts of the COVID-19 pandemic, (4) comparing the COVID-19 pandemic with the previous public health crises, (5) measuring impacts of the COVID-19 pandemic, and (6) proposing different remedial and management actions (discussing resumption of activities). These papers focused on different aspects of the hospitality industry, including the recovery of the industry after the pandemic, market demands, revenue losses, the COVID-19 spreading patterns in the industry, job losses, safety and health, employment issues of hospitality workers, travel behavior, preference of customers and social costs. The reviewed papers used a variety of research methodologies, such as the SEIR model, epidemiological model, agent-based model, supply and demand curve, DSGE model, crowd flow simulation model, secondary data analysis, seasonal autoregressive integrated moving average model, scenario analysis, trend analysis, descriptive analysis, contingent valuation model, content analysis, and analyzing questionnaire data.

In general, conducting a systematic literature review has several limitations. The first limitation is identifying and analyzing papers published in a specific time frame. The second limitation is the inability to discover individual relevant papers arising from a limited number of keywords. The third limitation is using a limited number of search databases for article discovery.

Although we defined several search keywords and followed PRISMA guidelines, it is possible that some papers that met the inclusion criteria were not considered in our review. We did not include the papers published after August 2020 as several papers on the topic have just started to emerge. Second, we selected papers only from Web of Science, Science Direct, and Google Scholar databases. Third, we could not include articles where authors investigated the hospitality industry at the time of the COVID-19 pandemic without mentioning the hospitality industry, tourism industry, event industry, hotel industry, and restaurant industry. Fourth, one of the main challenges of this review was defining inclusion criteria. Because of the very timely issue of the COVID-19 pandemic, we defined broad inclusion criteria. Therefore, we could not include several studies that met inclusion criteria but generated by institutions outside of the traditional academic publishing and distribution channels.

Despite the above limitations, we identified the hospitality industry’s main aspects in the face of the COVID-19 pandemic. These include the recovery of the hospitality industry (discussed by 19% of included papers), market demand (18% of papers), revenue losses (16% of papers), the COVID-19 spreading patterns in the industry (14% of papers), job losses (10% of included papers), safety and health aspects (8% of papers), issues related to the employment of hospitality workforce (7% of papers), travel behaviors (4% of papers), preferences of customers (2% of papers), and social costs of pandemic discussed by 2% of included papers.

It should be noted that there are numerous other fertile research areas and methodologies that can be applied by multidisciplinary research teams to study the effects of the COVID-19 pandemic on the hospitality industry. Such approaches and methods include (1) using complex system science frameworks such as syndemics, (2) developing simulation modeling in different types of system dynamics, discrete event simulation, agent-based modeling, and Monte Carlo/risk analysis simulation, (3) investigating the application of new technologies such as educational technology, information technology, and robotics in response to the pandemic, (4) using artificial intelligence in different types of machine learning, deep learning and neural networks, and (5) developing the best practices concerning the pandemic (see Figure 8). These research approaches can be used to analyze the main aspects of the hospitality industry at the time of the COVID-19 pandemic, such as developing sustainable industry, recovery and resilience of the hospitality industry, the safety of customers, issues of undocumented workers, market demand, and emerging the new market, hostile behavior toward customers, and the risks of resumption of activities during the pandemic.

For example, due to the complex and dynamic nature of the current pandemic, the use of a wide array of complex systems science frameworks and simulation modeling can make an important contribution by examining how the synergistic effects of work and living conditions, as well as COVID-19 government and corporate responses, can influence the long-term health and safety of tourism and hospitality workers. Along these lines, the development and application of new technologies and equipment in the hospitality industry should protect guests and workers alike. Finally, other potential areas of research include the use of machine learning and artificial intelligence in building a more sustainable tourism and hospitality industry and developing the best practices in improving the industry’s resilience in the future.

## Figures and Tables

**Figure 1 ijerph-17-07366-f001:**
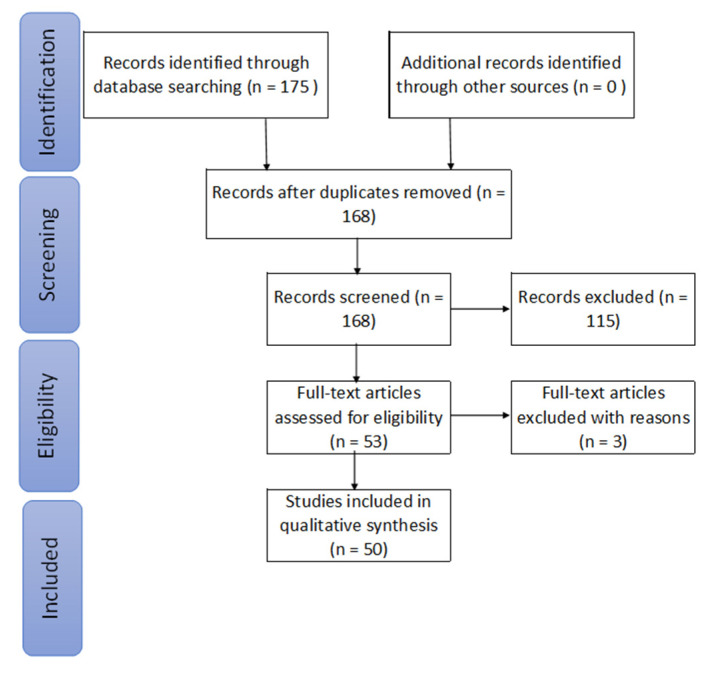
Chart of the selection strategy following PRISMA guidelines [13].

**Figure 2 ijerph-17-07366-f002:**
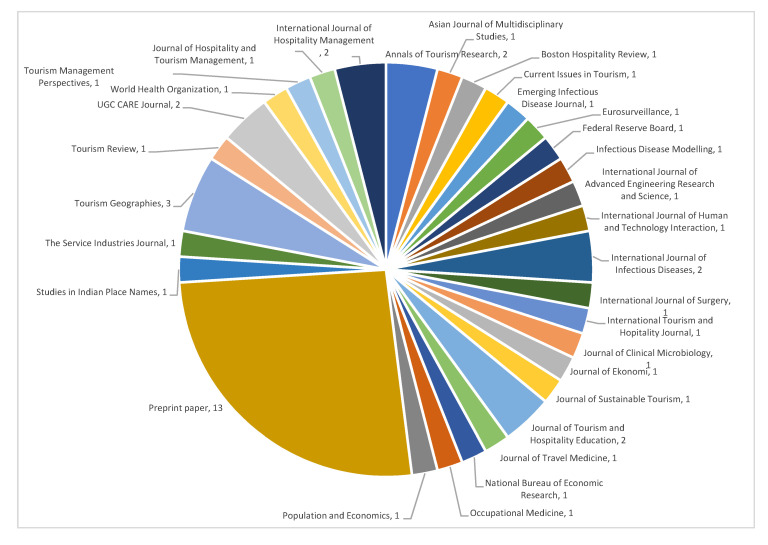
Publication source among included papers.

**Figure 3 ijerph-17-07366-f003:**
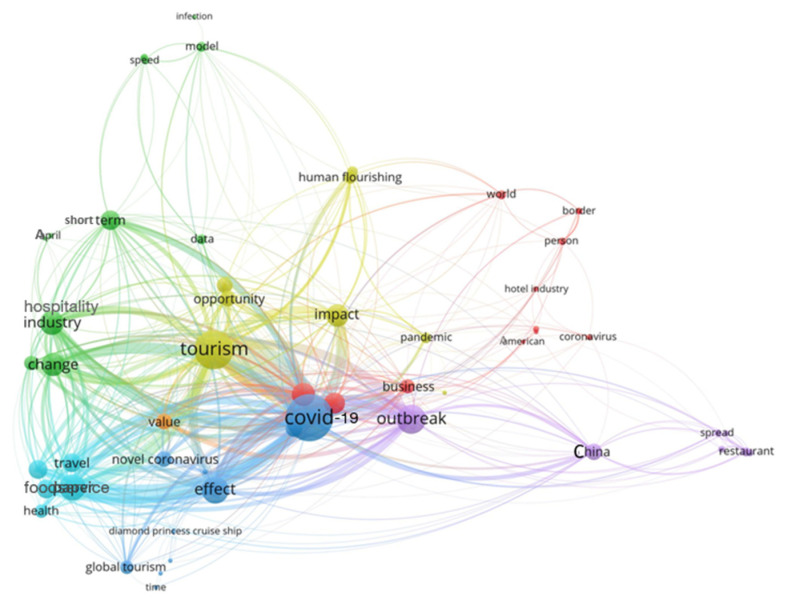
The map of the co-occurrence of the terms in the title and abstract of recorded papers.

**Figure 4 ijerph-17-07366-f004:**
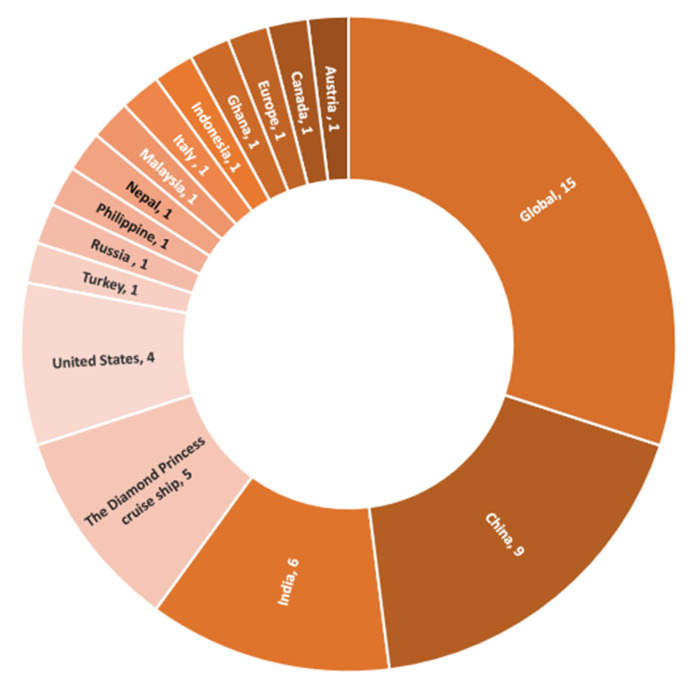
Geographic location among recorded papers.

**Figure 5 ijerph-17-07366-f005:**
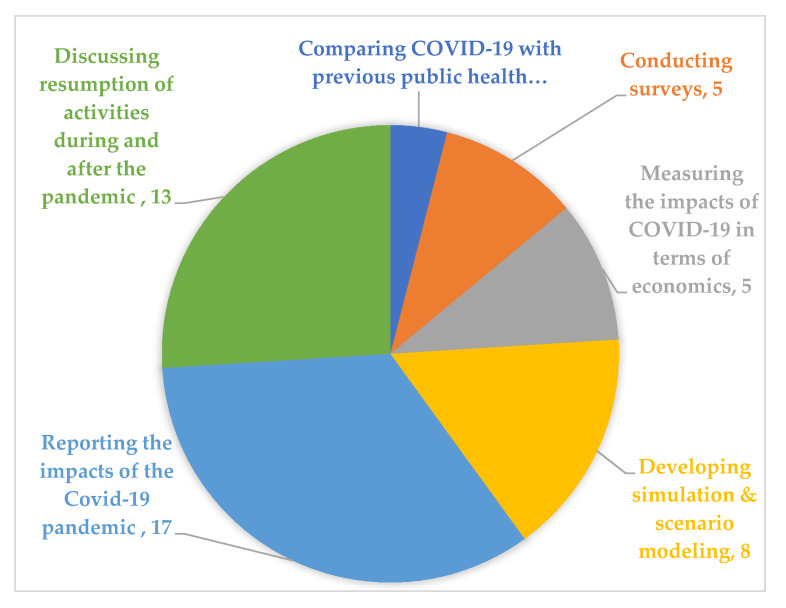
Research approach among included papers.

**Figure 6 ijerph-17-07366-f006:**
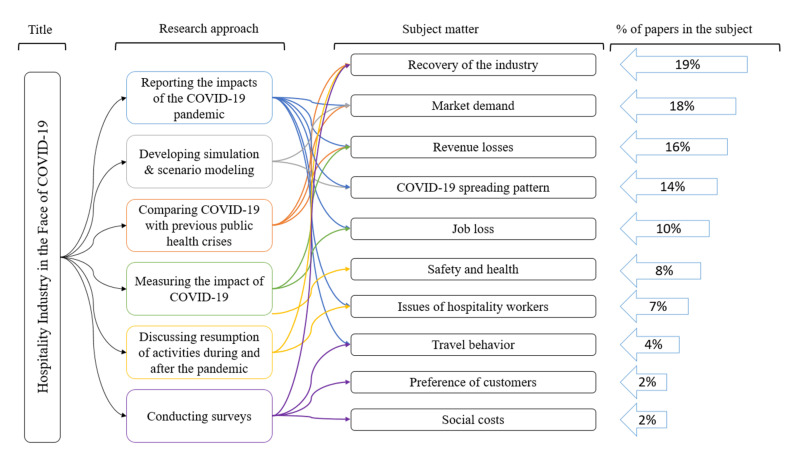
Research aspects of the hospitality industry among included papers.

**Figure 7 ijerph-17-07366-f007:**
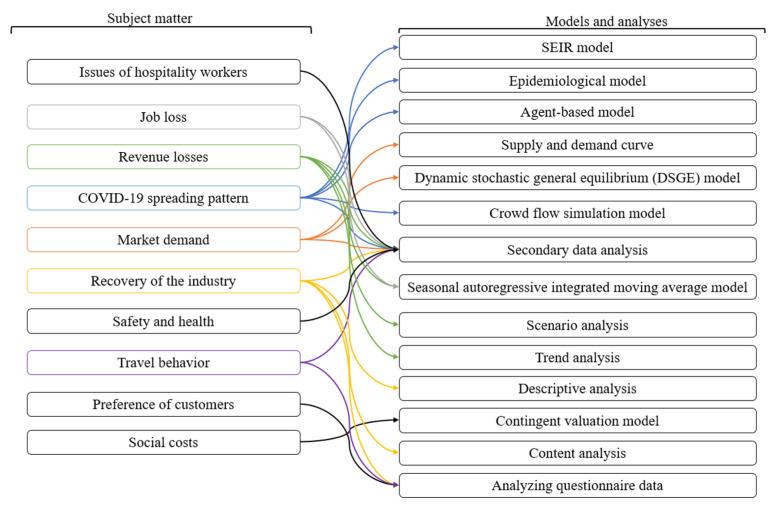
Research models and analyses of included papers.

**Figure 8 ijerph-17-07366-f008:**
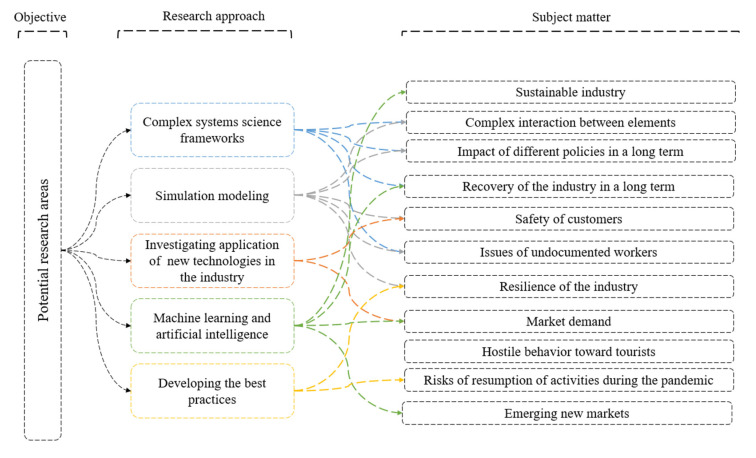
Future research approaches to study the hospitality industry in the face of COVID-19.

**Table 1 ijerph-17-07366-t001:** Keywords used in the literature search.

Row	Keywords
Search 1	COVID-19 AND hospitality industry
Search 2	COVID-19 AND event industry
Search 3	COVID-19 AND hotel industry
Search 4	COVID-19 AND restaurant industry
Search 5	COVID-19 AND tourism industry

**Table 2 ijerph-17-07366-t002:** Characteristics of the included papers.

Reference	Title	Segment of Industry	Geographic Location	Approach
[21]	Pandemics, tourism and global change: A rapid assessment of COVID-19	Airlines, Accommodation, sports events, restaurants, cruises	Global	Comparing COVID-19 with previous public health crises
[22]	Hedging feasibility perspectives against the COVID-19 for the international tourism sector	Tourism expenditure, inbound and outbound tourism, conference tourism, pilgrimage tourism, virtual reality tourism	Global	Comparing COVID-19 with previous public health crises
[23]	How are small businesses adjusting to COVID-19? Early evidence from a survey	Restaurant industry	United States	Conducting surveys
[24]	COVID-19 crises and tourist travel risk perceptions	Inbound and outbound tourism, tourist’s hygiene and safety	China	Conducting surveys
[25]	Social costs of tourism during the COVID-19 pandemic	The social cost of tourism	China	Conducting surveys
[26]	Effect of COVID-19 on preference for private dining facilities in restaurants	Private dining facilities in restaurants	United States	Conducting surveys
[27]	COVID-19, aftermath, impacts, and hospitality firms: An international perspective	Hospitality firms including hotels, restaurants, bars, winery, and agritourism	Global	Conducting surveys
[28]	Tracking labor market developments during the COVID-19 pandemic: A preliminary assessment	Hospitality job loss	United States	Measuring the impact of COVID-19
[29]	how much did the tourism industry lost? estimating earning loss of tourism in the Philippines	Foreign visitor arrivals	Philippine	Measuring the impact of COVID-19
[30]	Assessing the Short-term Impacts of COVID-19 Pandemic on Foreign Visitor’s Demand for Turkey: A Scenario Analysis	Foreign Visitor Arrivals and Foreign Visitor’s Demand	Turkey	Measuring the impact of COVID-19
[31]	COVID-19: A nightmare for the Indian economy	Hospitality job loss	India	Measuring the impact of COVID-19
[32]	A Survey on some of the global effects of the COVID-19 pandemic	Hotel industry, aviation industry	Global	Measuring the impact of COVID-19
[33]	How many infections of COVID-19 there will be in the “diamond princess“-predicted by a virus transmission model based on the simulation of crowd flow	Cruise industry	The Diamond Princess cruise ship	Developing simulation & scenario modeling
[34]	Transmission potential of the novel coronavirus (COVID-19) onboard the diamond Princess Cruises Ship,2020	Cruise industry	The Diamond Princess cruise ship	Developing simulation & scenario modeling
[35]	Estimating the asymptomatic proportion of coronavirus disease 2019 (COVID-19) cases on board the Diamond Princess cruise ship, Yokohama, Japan, 2020	Cruise industry	The Diamond Princess cruise ship	Developing simulation & scenario modeling
[36]	COVID-19 outbreak on the Diamond Princess cruise ship: Estimating the epidemic potential and effectiveness of public health countermeasures	Cruise industry	The Diamond Princess cruise ship	Developing simulation & scenario modeling
[37]	Estimation of the reproductive number of novel coronavirus (COVID-19) and the probable outbreak size on the Diamond Princess cruise ship: A data-driven analysis	Cruise industry	The Diamond Princess cruise ship	Developing simulation & scenario modeling
[1]	Effect of Coronavirus disease (COVID-19) to the tourism industry	Supply-demand in tourism industry	Global	Developing simulation & scenario modeling
[38]	Sustainable and resilient strategies for touristic cities against COVID-19: An agent-based approach	Touristic cities	Italy	Developing simulation & scenario modeling
[2]	Coronavirus pandemic and tourism: Dynamic stochastic general equilibrium modeling of infectious disease outbreak	Tourism demand	Global	Developing simulation & scenario modeling
[39]	A pandemic in times of global tourism: Superspreading and exportation of COVID-19 cases from a ski area in Austria	Tourists location	Austria	Reporting the impacts of the COVID -19 pandemic
[40]	COVID-19 outbreak associated with air conditioning in restaurant, Guangzhou, China, 2020	Restaurant industry	China	Reporting the impacts of the COVID-19 pandemic
[41]	Expected tourist attractions after pandemic COVID-19	Tourists locations	China	Reporting the impacts of the COVID-19 pandemic
[42]	A content analysis of Chinese news coverage on COVID-19 and tourism	Different aspects of tourism industry	China	Reporting the impacts of the COVID-19 pandemic
[43]	The effect of Coronavirus (COVID-19) in the tourism industry in China	Inbound and outbound flights, hotel industry, restaurant industry	China	Reporting the impacts of the COVID-19 pandemic
[44]	Airbnb, COVID-19 risk and lockdowns: global evidence	Hotel industry	China	Reporting the impacts of the COVID-19 pandemic
[45]	Impact of Covid-19 pandemic on the tourism sector	Tourism sectors	India	Reporting the impacts of the COVID-19 pandemic
[46]	The Movement Control Order (MCO) for COVID-19 crisis and its impact on tourism and hospitality sector in Malaysia	Tourists arrivals	Malaysia	Reporting the impacts of the COVID-19 pandemic
[47]	Effects of Coronavirus disease (COVID-19) on tourism industry of India	Tourism sectors	India	Reporting the impacts of the COVID-19 pandemic
[48]	Indian tourism industry and COVID-19: Present scenario	Tourism sectors	India	Reporting the impacts of the COVID-19 pandemic
[49]	The socio-economic implications of the Coronavirus and COVID-19 pandemic: A review	Hotel industry, aviation, sports industry, travel supply and demand	Global	Reporting the impacts of the COVID-19 pandemic
[50]	health and safety considerations for hotel cleaners during COVID-19	Hotel industry	Global	Reporting the impacts of the COVID-19 pandemic
[51]	Spillover of COVID-19: impact on the global economy	Travel industry, restaurant industry, sports industry, event industry, entertainment industry	Global	Reporting the impacts of the COVID-19 pandemic
[52]	An initial assessment of economic impacts and operational challenges for the tourism & hospitality industry due to COVID-19	Tourism-related businesses	Ghana	Reporting the impacts of the COVID-19 pandemic
[53]	A preliminary study of novel Coronavirus disease (COVID-19) outbreak: A pandemic leading crisis in tourism industry of Nepal	Tourism-related businesses	Nepal	Reporting the impacts of the COVID-19 pandemic
[54]	COVID-19: potential effects on Chinese citizens’ lifestyle and travel	Tourists’ behavior and preferences	China	Reporting the impacts of the COVID-19 pandemic
[55]	COVID-19 and undeclared work: Impacts and policy responses in Europe	Hospitality workforce	Europe	Reporting the impacts of the COVID-19 pandemic
[56]	Revitalization strategy for small and medium enterprises after Corona Virus disease pandemic (COVID-19) in Yogyakarta	Tourism-related businesses	Indonesia	Discussing resumption of activities during and after the pandemic
[57]	Socialising tourism for social and ecological justice after COVID-19	Socializing tourism	Global	Discussing resumption of activities during and after the pandemic
[58]	Effect of COVID-19 on restaurant industry–How to cope with changing demand	Restaurant industry	India	Discussing resumption of activities during and after the pandemic
[59]	Would hotel industry have to redo the rooms/housekeeping standards post COVID? Instilling greater confidence to bring back the customers	Hotel industry	India	Discussing resumption of activities during and after the pandemic
[60]	Reconnecting tourism after COVID-19: The paradox of alterity in tourism areas	6 foot-tourism	Canada	Discussing resumption of activities during and after the pandemic
[61]	A post-COVID-19 model of tourism and hospitality workforce resilience	Hospitality workforce	Global	Discussing resumption of activities during and after the pandemic
[62]	COVID-19–We urgently need to start developing an exit strategy	Hotels and aviation industries	Global	Discussing resumption of activities during and after the pandemic
[63]	Coronavirus and tourism	Behavior and preferences of tourists	Russia	Discussing resumption of activities during and after the pandemic
[64]	Operational considerations for COVID-19 management in the accommodation sector: Interim guidance, 31 March 2020	Hotel industry	Global	Discussing resumption of activities during and after the pandemic
[65]	COVID-19 and China’s hotel industry: Impacts, a disaster management framework, and post-pandemic agenda	Hotel industry	China	Discussing resumption of activities during and after the pandemic
[66]	Understanding the effects of COVID-19 on the health and safety of immigrant hospitality workers in the United States	Hospitality workforce	United States	Discussing resumption of activities during and after the pandemic
[67]	Reset redux: Possible evolutionary pathways towards the transformation of tourism in a COVID-19 world	Transformation of tourism	Global	Discussing resumption of activities during and after the pandemic
[68]	Navigating hotel operations in times of COVID-19	Hotel industry	Global	Discussing resumption of activities during and after the pandemic

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
