# Peer review of "The Hospitality Industry in the Face of the COVID-19 Pandemic: Current Topics and Research Methods"

_ijerph, 2020, doi:10.3390/ijerph17207366_

Round 1

Reviewer 1 Report

The literature review is well-structured and up to date. It enables the reader to grasp the main themes in the current literature discussion on COVID impacts on hospitality and tourism industry and identify future research topics.

There are few things that need some clarification.

First, the title contains the term "hospitality industry". Then, when writing about the structure of reviewed papers by segment (line 112) you write "papers mainly focused on the tourism industry, followed by the hospitality industry", which suggests that both "hospitality industry" and "tourism industry" are parts of "hospitality industry". Obviously, the terms "hospitality industry", "tourism industry", "travel industry" are related to each other and understood differently in different countries, but when reporting such structure by segment I would like to see more concrete difivision (e.g. hotel industry, restaurant industry, travel agencies, transportation, parks & recreation, and maybe a broad category "unspecified"). In the current form figure 4 does not bring much useful information.

Second, the inclusion criteria for the review (line 83-83) are not very precise. Did you take into account only papers published in peer-reviewed academic publications, or also "grey literature"? In figure 2 I see academic journals, but also probably reports (e.g. Federal Reserve Board, National Bureau of Economic Research) and pre-prints which were not yet peer-reviewed and published (c. 1/4 of all). I see a point of using such an approach (very timely issue, traditional publication process takes time), but it should be maybe more clearly described why you used these sources, but not e.g. book chapters.

Minor issues:

Line 35 - "difficulty breathing" should rather be "breathing problems"?

Table 2 - maybe consider adding paper titles to the table apart from reference numbers - it would make the table more readible.

Line 367 - question mark probably unnecessary.

Author Response

Reviewer

No of comment

Comments

Revision

1

1

First, the title contains the term "hospitality industry". Then, when writing about the structure of reviewed papers by segment (line 112) you write "papers mainly focused on the tourism industry, followed by the hospitality industry", which suggests that both "hospitality industry" and "tourism industry" are parts of "hospitality industry". Obviously, the terms "hospitality industry", "tourism industry", "travel industry" are related to each other and understood differently in different countries, but when reporting such structure by segment I would like to see more concrete difivision (e.g. hotel industry, restaurant industry, travel agencies, transportation, parks & recreation, and maybe a broad category "unspecified"). In the current form figure 4 does not bring much useful information.

Thank you very much for your comment.

We have clarified this point as follows.

Comment: p.7 line 112-113

Revision: p.4-8 line 130-131

To clarify, we added specific aspects of the hospitality industry, such as the hotel industry, restaurant industry, travel agencies, or even transportation to Table 2 in the “segment of industry” column. Also, we removed Figure 4 and line 112 to avoid any misunderstanding..

1

2

Second, the inclusion criteria for the review (line 83-83) are not very precise. Did you take into account only papers published in peer-reviewed academic publications, or also "grey literature"? In figure 2 I see academic journals, but also probably reports (e.g. Federal Reserve Board, National Bureau of Economic Research) and pre-prints which were not yet peer-reviewed and published (c. 1/4 of all). I see a point of using such an approach (very timely issue, traditional publication process takes time), but it should be maybe more clearly described why you used these sources, but not e.g. book chapters.

Thank you very much for your comment.

We have clarified this point as follows.

Comment: p.2 line 83-85

Revision: p.3 line 105-108

We updated the search strategy and conducted article discovery again based on the following statement:

 “Because of the very timely issue of the COVID-19 pandemic, we included documents in the forms of peer-reviewed academic publications, grey literature, and pre-print articles. However, we excluded secondary sources that were not free or open access, letters, newspaper articles, viewpoints, presentations, anecdotes, and posters.”

However, we could not find any other articles. Also, we fully explained the limitations of the search strategy in the Conclusion section. 

Revision: p.17-18 line 474-495

1

3

Line 35 - "difficulty breathing" should rather be "breathing problems"?

Thank you very much for your comment.

We have clarified this point as follows.

Comment: p.1 line 35

Revision: p.1 line 34

1

4

Table 2 - maybe consider adding paper titles to the table apart from reference numbers - it would make the table more readible.

Thank you very much for your comment.

We have clarified this point as follows.

Comment: p.3-6 line 106-107

Revision: p.4-8 line 130-131

1

5

Line 367 - question mark probably unnecessary.

Thank you very much for your comment.

We have clarified this point as follows.

Comment: p.13 line 367

Revision: p.15 line 396

Reviewer 2 Report

Dear Authors,

I recommend the paper for publication after some minor revisions, namely:

- The Introduction section seems to be limited and have a limited literature review in the field.

- The Methodology section should be rooted in the research theory.

- The Results section lacks the in depth interpretation of the Figures and Tables.

- Rethink the “Conclusion” section - try to:

    1. highlight key findings in your “Results” section,
    2. place the paper within the context of how your research advances  research about the topic,
    3. describe how a previously identified gap in the literature (your Introduction section) has been filled by your research,
    4. demonstrate the importance of your recommendations/suggestions,
    5. define the limitations of your research,
    6. propose possible new or expanded ways of thinking about the research problem referring to the Figure 9.

I do hope you find the comments helpful as you move forward with your paper.

Author Response

Reviewer

No of comment

Comments

Revision

2

6

The Introduction section seems to be limited and have a limited literature review in the field.

Thank you very much for your comment.

We have clarified this point as follows.

Comment: p.1-2 line 30-60

Revision: p.1-3 line 29-84

2

7

The Methodology section should be rooted in the research theory.

Thank you very much for your comment.

We have clarified this point as follows.

Comment: p.2 line 83-85

Revision: p.3 line 105-108

We updated the search strategy and conducted article discovery again based on the following statement:

“Because of the very timely issue of the COVID-19 pandemic, we included documents in the forms of peer-reviewed academic publications, grey literature, and pre-print articles. However, we excluded secondary sources that were not free or open access, letters, newspaper articles, viewpoints, presentations, anecdotes, and posters.”

However, We could not find any other article. Also, we fully explained the limitations of the search strategy in the Conclusion section.Revision: p.17-18 line 474-495

2

8

The Results section lacks the in depth interpretation of the Figures and Tables.

Thank you very much for your comment.

We have clarified this point as follows.

Comment: p.3-8 line 102-127

Revision: p.8-10 line 134-155

We added more descriptions to all figures in the results section.

2

9

- Rethink the “Conclusion” section - try to:

·         highlight key findings in your “Results” section,

·         place the paper within the context of how your research advances  research about the topic,

·         describe how a previously identified gap in the literature (your Introduction section) has been filled by your research,

·         demonstrate the importance of your recommendations/suggestions,

·         define the limitations of your research,

·         propose possible new or expanded ways of thinking about the research problem referring to the Figure 9.

Thank you very much for your comment.

We have clarified this point as follows.

Comment: p.15-16 line 420-454

Revision: p.17-18 line 450-519

We totally re-organized the conclusion section to address all elements of this comment.

Reviewer 3 Report

The topic discussed is interesting and summarizing the state of the art is essential given the speed with which works related to the COVID-19 pandemic are coming out. But major improvements can be made:

  • It is striking that there is no literature review section that lists previous similar studies in the field of tourism and hospitality, even if it is not related to COVID-19, or related to studies generated in relation to COVID- 19, even if they have nothing to do with tourism and hospitality. A section of this type would be much needed to support the adequacy of the type of study and methodology.
  • The words used in the search (Table 1) have a problem. The word "industry" is used by some authors but not by all. Some consider that "industry" is the secondary sector and tourism and hospitality is the tertiary sector (services) and they do not use this term. Only some authors use this word to assimilate tourism to industry. This can greatly reduce the items your search finds. It would be necessary, as a minimum, to indicate this subject in limitations.
    Figures 2, 4, 5 and 6 are very visual but perhaps it would be better if these data were in a table, indicating frequencies and percentages.
  • It is missed that in the conclusions section there is not a subsections of study limitations, possible lines that should be investigated in relation to the review topic, etc.
  • There is also some formatting error, such as in the format of the citations on lines 216 and 317.

Once these aspects are corrected, the paper would be quite good.

Author Response

Reviewer

No of comment

Comments

Revision

3

10

It is striking that there is no literature review section that lists previous similar studies in the field of tourism and hospitality, even if it is not related to COVID-19, or related to studies generated in relation to COVID- 19, even if they have nothing to do with tourism and hospitality. A section of this type would be much needed to support the adequacy of the type of study and methodology.

Thank you very much for your comment.

We have clarified this point as follows.

Revision: p.2 line 59-69

We added the following text

Although no study used a systematic literature review to investigate the hospitality industry in the face of the COVID-19 pandemic, conducting the systematic review is common in the context of hospitality. For example, Yu et al. [14] conducted a comprehensive review of abusive supervision in hospitality and tourism. Gorska-Warsewicz & Kulykovets [15] conducted a systematic literature review and selected 26 studies to analyze hotel brand loyalty. The study used the Joanna Briggs Institute's critical appraisal checklist to address the risk of bias among the included records. Sharma et al. [16] used a systematic review and analyzed 403 published papers in 13 established hospitality journals to address green hospitality practices. The study proposed a unified conceptual framework based on discovering seven research areas under eco-innovative procedures. Chi et al. [17] discussed applying artificial intelligence in the hospitality industry, specifically service delivery. The study reviewed 63 publications and identified seven major themes.

3

11

The words used in the search (Table 1) have a problem. The word "industry" is used by some authors but not by all. Some consider that "industry" is the secondary sector and tourism and hospitality is the tertiary sector (services) and they do not use this term. Only some authors use this word to assimilate tourism to industry. This can greatly reduce the items your search finds. It would be necessary, as a minimum, to indicate this subject in limitations.

Thank you very much for your comment.

We have clarified this point as follows.

Comment: p.2 line 75

Revision: p.3 line 99

We thoroughly discussed the limitations of the search strategy in the conclusion section.

Revision: p.17-18 line 474-488

3

12

Figures 2, 4, 5 and 6 are very visual but perhaps it would be better if these data were in a table, indicating frequencies and percentages.

Thank you very much for your comment.

We have clarified this point as follows.

Comment: p.6-8 line 107-126

Revision: p.8-10 line 134-155

We added percent and number to the figures to improve them. Also, we added more descriptions for each Figure,

3

13

It is missed that in the conclusions section there is not a subsections of study limitations, possible lines that should be investigated in relation to the review topic, etc.

Thank you very much for your comment.

We have clarified this point as follows.

We fully explained the limitations of the review in the Conclusion section.

Revision: p.17-18 line 474-488

3

14

There is also some formatting error, such as in the format of the citations on lines 216 and 317.

Thank you very much for your comment.

We have clarified this point as follows.

Comment: p.10,12 line 216, 317

Revision: p.12,14 line 245,346

Round 2

Reviewer 3 Report

The above problems have been remedied reasonably well.